# Differential Susceptibility of Retinal Neurons to the Loss of Mitochondrial Biogenesis Factor Nrf1

**DOI:** 10.3390/cells11142203

**Published:** 2022-07-14

**Authors:** Takae Kiyama, Ching-Kang Chen, Annie Zhang, Chai-An Mao

**Affiliations:** 1Ruiz Department of Ophthalmology and Visual Science, McGovern Medical School at The University of Texas Health Science Center at Houston (UTHealth), 6431 Fannin St., MSB 7.024, Houston, TX 77030, USA; takae.kiyama@uth.tmc.edu (T.K.); annie.zhang@uth.tmc.edu (A.Z.); 2Department of Ophthalmology, Baylor College of Medicine, One Baylor Plaza, Houston, TX 77030, USA; ching-kang.chen@bcm.edu; 3The MD Anderson Cancer Center/UTHealth Graduate School of Biomedical Sciences, Houston, TX 77030, USA

**Keywords:** mitochondrial biogenesis, Nrf1, photoreceptor degeneration, bipolar cells, ganglion cells, transcriptome, RNA-seq, retina disease

## Abstract

The retina, the accessible part of the central nervous system, has served as a model system to study the relationship between energy utilization and metabolite supply. When the metabolite supply cannot match the energy demand, retinal neurons are at risk of death. As the powerhouse of eukaryotic cells, mitochondria play a pivotal role in generating ATP, produce precursors for macromolecules, maintain the redox homeostasis, and function as waste management centers for various types of metabolic intermediates. Mitochondrial dysfunction has been implicated in the pathologies of a number of degenerative retinal diseases. It is well known that photoreceptors are particularly vulnerable to mutations affecting mitochondrial function due to their high energy demand and susceptibility to oxidative stress. However, it is unclear how defective mitochondria affect other retinal neurons. Nuclear respiratory factor 1 (Nrf1) is the major transcriptional regulator of mitochondrial biogenesis, and loss of *Nrf1* leads to defective mitochondria biogenesis and eventually cell death. Here, we investigated how different retinal neurons respond to the loss of *Nrf1*. We provide in vivo evidence that the disruption of *Nrf1*-mediated mitochondrial biogenesis results in a slow, progressive degeneration of all retinal cell types examined, although they present different sensitivity to the deletion of *Nrf1*, which implicates differential energy demand and utilization, as well as tolerance to mitochondria defects in different neuronal cells. Furthermore, transcriptome analysis on rod-specific *Nrf1* deletion uncovered a previously unknown role of Nrf1 in maintaining genome stability.

## 1. Introduction

Mitochondria biogenesis is a subcellular process through which mitochondria are replenished and remodeled through continuously importing and incorporating new proteins and lipids, replicating mitochondrial DNA, transcribing mitochondrion-encoded genes, and fusing and dividing in response to cellular demands and bioenergetics loads. This process is essential for maintaining a healthy and functional mitochondria network for energy metabolism, calcium homeostasis, biosynthesis of amino acids, cholesterol, and phospholipids, eliminating excessive reactive oxygen species, and apoptosis [1,2,3,4,5,6].

A number of transcriptional regulators, including peroxisome proliferative activated receptor gamma coactivator 1 (PGC-1) family members, nuclear respiratory factor 1 (Nrf1), and nuclear respiratory factor 2 (Nrf2/GABP), have been identified as key regulators for mitochondria biogenesis in different contexts [7,8,9,10]. Among them, Nrf1 is identified as an evolutionarily conserved transcription activator that binds to GC-rich DNA elements in promoters of a host of nuclear genes encoding proteins involved in mitochondrial structure and functions [11,12,13,14,15]. Multiple studies using ChIP-seq analysis have identified distinct sets of Nrf1’s target genes in different cell types, suggesting that Nrf1 acts in a context-dependent manner in regulating cell growth, differentiation, and mitochondrial biogenesis [16,17,18,19]. In addition, Nrf1 interacts with other proteins involved in different cellular functions. For example, it interacts directly with Auts2 to confer transcriptional activity for non-canonical polycomb repressive complex ncPRC1.3 [17]. It also interacts with poly(ADP-ribose) polymerase 1 (PARP-1) and PARP-1 PARylates DNA-binding domain of Nrf1 for transcriptional regulation [20]. Post-translational modifications on various Nrf1 domains may in addition regulate transcriptional activity in different cellular contexts [21].

Previously, we have demonstrated that defective *Nrf1*-mediated mitochondrial biogenesis significantly affects cell proliferation, migration, and metabolic pathways in retinal progenitor cells (RPCs) during retinal development and uncovered essential roles of *Nrf1* in the survival of newly specified retinal ganglion cells (RGCs) and neurite outgrowth in retinal explants. In rod photoreceptor cells (PRs), *Nrf1* deletion causes defective mitochondrial morphology, position, and function, leading to a slow, progressive degeneration of all rod PRs after 4–5 months of age. In a separate study, we further found that when *Nrf1* is deleted in a subset of Tbr1-expressing OFF-RGCs [22], these RGCs also displayed a similar pattern of slow degeneration [17]. In the same study, we have also observed the reduction of Tbr1-expressing hippocampal neurons.

In the current study, we expand previous studies by covering additional types of retinal neurons and find in them differential sensitivity to *Nrf1* deletion. Our findings provide in vivo evidence that *Nrf1*-mediated pathways have cell-type-specific functions in different retinal neural types. Through transcriptomic analyses, we further discovered a small subset of early responsive genes in *Nrf1* deleted rod PRs and uncovered a novel role of Nrf1 in maintaining genome stability.

## 2. Materials and Methods

### 2.1. Animal Breeding

*Nrf1^flox^*, *Rho^iCre^*, *Opn4^cre^*, *HRGP-Cre*, and *Pcp2-Cre* mice were described previously [23,24,25,26,27,28]. We have confirmed that the expression of Cre recombinase in these lines do not contribute to the cell death phenotype described in this study. All animal procedures followed the US Public Health Service Policy on Humane Care and Use of Laboratory Animals and were approved by the Animal Welfare Committee at The University of Texas Health Science Center at Houston and the Animal Welfare Committee at the Baylor College of Medicine.

### 2.2. Immunohistochemistry

Flat-mounted retinas were fixed with 4% paraformaldehyde (PFA; Electron Microscopy Sciences). Retinas were then embedded in 4% low melting agarose and sectioned into 100 μm thickness with Leica VT1000S vibratome (Leica Biosystems, Deer Park, IL, USA). Retinal sections or flat-mounted retinas were incubated with the primary antibodies for 3 days at 4 °C. The primary antibodies used were rabbit anti-R/G opsin (1:1000, catalog #AB5405, Sigma), rabbit anti-B opsin (1:1000, catalog #AB5407, Sigma, St. Louis, MO, USA), rabbit anti-melanopsin (1:1000, catalog #AB-N38, Advanced Targeting Systems, Carlsbad, CA, USA), chicken anti-GFP (1:1000, catalog #A10262, Thermo Fisher Scientific, Waltham, MA, USA), rabbit anti-PKCα (1:600, catalog #P4334, Sigma), mouse anti-Syt2 (1:1000, catalog #Znp-1, DSHB, The University of Iowa, Iowa City, IA, USA), and mouse anti-Nrf1 (1:300, catalog #PCRP-NFR1-3D4; DSHB). Secondary antibodies conjugated with Alexa-488 and -555 (Thermo Fisher Scientific) were used at 1:800 dilution. DAPI (2.5 μg/mL, catalog #D1306; Thermo Fisher Scientific) was used to stain nuclei. Images were captured using a Zeiss LSM 780 confocal microscope (Carl Zeiss, White Plains, NY, USA) and exported as TIFF files into Adobe Photoshop (Adobe Systems, San Jose, CA, USA). Cell counting was conducted using the cell counter plugin in the NIH ImageJ (NIH, Bethesda, MD, USA).

### 2.3. Photopic Electroretinography

Photopic ERG recordings were performed on mice dark-adapted for two hours. Briefly, mice were anesthetized with ketamine/xylazine (150/10 mg/kg; ip), and the pupils were dilated for 10 min with sequential topical eye drops of 1% tropicamide and 2.5% phenylephrine (Bausch & Lomb, Tampa, FL, USA). A total of 5 μL of filtered 1X PBS was added to the surface of the eyes to prevent corneal clouding and cataract formation. Body temperature was monitored by a rectal probe (Braintree Scientific, Braintree, MA, USA) and maintained at 35 °C to 37 °C using a plastic heating coil with 43 °C circulating water. ERG from both eyes was recorded simultaneously (UTAS BigShot^TM^ system; LKC Technologies, Gaithersburg, MD, USA) after exposure to a background illumination of 30 cd m^−2^ white light for 10 min. Typically, the difference in the photopic b-wave amplitudes between the two eyes were <10%. Averaged responses to 90 flashes of the intensity of 25 cd s m^−2^ delivered at 1 Hz were recorded. A typical recording session lasted about 20 min for each animal.

### 2.4. RNA-Sequencing Analysis

Four retinas isolated from male or female *wild-type* (*WT*) and *Nrf1^f/f^;Rho^iCre^* at 4 weeks old were pooled in the RNA later solution (Qiagen, Germantown, MD, USA). Three sets of pooled retinas were independently used for RNA-seq. Male and female retinas were sequenced separately to determine whether gender is a determining biological variable. Total RNA was extracted using an RNeasy mini kit (Qiagen, Germantown, MD, USA), and RNA sequencing was performed by Novogene (Sacramento, CA, USA). RNA libraries were prepared using standard Illumina protocols. Paired-end reads (150 nt) were obtained using an Illumina Novaseq 6000 Next Generation Sequencing instrument. Sequence reads were trimmed free of adaptor sequences and masked for low-complexity or low-quality sequences, then mapped to the mouse mm10 whole genome using the STAR (ver2.5) software (https://github.com/alexdobin/STAR). HITSeq v0.6.1 was used to count the read numbers mapped to each gene. FPKM of each gene was calculated based on the length of the gene and the read count mapped to this gene. Differential expression analyses of two groups were performed using the DESeq2 R package (2_1.6.3) (https://bioconductor.org/packages/release/bioc/html/DESeq2.html). The resulting *p*-values were adjusted using Benjamini and Hochberg’s approach for controlling the false discovery rate. *Nrf1*-dependent genes (fold change > 1.6667; adjusted *p*-value < 0.05) were analyzed by using the PANTHER Overrepresentation Test (Released 20220202). The raw datasets and normalized count data for each sample have been deposited in NCBI (Geo dataset: GSE150258).

### 2.5. Quantitative Reverse Transcriptase PCR

Two retinas from *WT* or *Nrf1^f/f^*;*Rho^iCre^* of multiple littermates at 4 weeks old were extracted using TRI reagent (MilliporeSigma, Burlington, MA, USA). First-strand cDNA was synthesized using iScript Reverse Transcription Super Mix for RT-qPCR (Bio-Rad, Hercules, CA, USA). Real-time PCR was performed using the CFX Connect Real-Time System (Bio-Rad, Hercules, CA, USA) with iTaq Universal SYBR Green Supermix (Bio-Rad, Hercules, CA, USA). Relative RNA levels were normalized to that of *β-actin*.

### 2.6. Terminal Deoxynucleotidyl Transferase dUTP Nick-End Labeling (TUNEL) Assay

Eye balls were extracted and fixed with 4% PFA for 1.5 hrs, and washed with PBS and cryo-sectioned into 30 μm thickness. Sections were washed with PBS then proceeded to TUNEL assay using the “In Situ Cell Death Detection Kit” (Roche Molecular Systems, Pleasanton, CA, USA). DAPI was used to stain nuclei.

### 2.7. Statistical Analysis

All data are presented as mean ± SD for each genotype. For all comparisons between genotypes, a two-tailed, two-sample Student’s *t*-test was used for all measurements and conducted in Microsoft Excel (Redmond, WA, USA). Results were considered significant when *p* < 0.05.

## 3. Results

To determine how different retinal neurons respond to the loss of *Nrf1*, we generated several cell type-specific *Nrf1* conditional knockouts by breeding *Nrf1^flox^* allele [29] to *HRGP-Cre*, *Opn4^Cre^*, or *Pcp2-Cre* to delete *Nrf1* in cone photoreceptors, intrinsically photosensitive retinal ganglion cells (ipRGCs), and a subset of cone and rod bipolar cells that express *Pcp2*, respectively.

### 3.1. Progressive Degeneration in Nrf1-Deficient Cone Photoreceptors

First, we examined the *Nrf1* deletion mediated by *HRGP-Cre* (*Nrf1^f/f^*;*HRGP-Cre*) (Figure 1A), a transgenic mouse line in which the *Cre* recombinase gene is driven by human red/green pigment (*HRGP*) promoter specifically in M-opsin-expressing cone photoreceptors [26]. The majority of cone photoreceptors co-express middle/long wavelength-sensitive opsin (M-opsin) and short wavelength-sensitive opsin (S-opsin) in mouse retina [30]; thus, *Nrf1* should be deleted in most of the cone photoreceptors in *Nrf1^f/f^;HRGP-Cre* retinas. We first examined the distribution of S-opsin^+^ or M-opsin^+^ cone photoreceptors by immuno-fluorescent staining using anti-blue opsin or anti-red/green opsin antibodies, respectively. In 3-month-old *Nrf1^f/+^* control retinas, S-opsin^+^ cone cells (S-cones) were distributed in a ventral-high, dorsal-low manner (Figure 1B), consistent with previous reports [30,31,32,33]. In contrast, M-opsin^+^ cone cells (M-cones) were distributed in a slightly dorsal-high and ventral-low manner in the control retina (Figure 1D), although this gradient is less apparent [30]. In *Nrf1^f/f^;HRGP-Cre* retinas, the expression of S-opsin and M-opsin were significantly reduced compared to *Nrf1^f/+^* retinas (Figure 1C,E). The reduction of S- and M-cones appears to be occurring uniformly in the entire retina without a noticeable sign of regional differences. Because of the severe reduction of S- and M-cones in 3-month-old *Nrf1^f/f^;HRGP-Cre* retinas, we next examined the functions of cone photoreceptors using photopic electroretinography (ERG). Photopic ERG b-wave amplitudes from light-adapted 4-week-old control and *Nrf1*-deficient mice were comparable (Figure 1F). At 6 weeks old, the photopic EGR b-wave of *Nrf1*-deficient mice started to decline gradually, and eventually reduced to ~10% that of the control mice at 14 weeks old (Figure 1F). ERG data indicate that deleting *Nrf1* in cone photoreceptors leads to the gradual loss of cone cells and cone-mediated function.

Next, we asked whether S- and M-cones respond to *Nrf1* deletion differently by examining them in 1- to 4-month-old *Nrf1^f/+^* and *Nrf1^f/f^;HRGP-Cre* retinas. Immunostaining images from different areas (dorsal and ventral for S-opsin; dorsal, ventral, nasal, and temporal for M-opsin) were counted. At 1 month old, there was no noticeable difference of S-cones between *Nrf1^f/+^* and *Nrf1^f/f^;HRGP-Cre* retinas, both in the dorsal and ventral areas (compare Figure 2A with Figure 2B, Figure 2G with Figure 2H, and Figure 2S with Figure 2T). At 2 months old, S-cones in the ventral retinas of *Nrf1*-deficient mice declined to ~40% that of the *Nrf1^f/+^* retina (Figure 2T), while in dorsal retinas, a slight, but less significant reduction of S-cones was observed (Figure 2S). At 3 months old, the number of S-cones in both dorsal and ventral retinas decreased to ~30–35% that of the *Nrf1^f/+^* retina (compare Figure 2C with Figure 2D, Figure 2I with Figure 2J, and Figure 2S with Figure 2T). At 4 months old, hardly any S-cones can be detected in both the dorsal and ventral areas of *Nrf1*-deficient retinas (compare Figure 2E with Figure 2F, Figure 2K with Figure 2L, and Figure 2S with Figure 2T).

M-cones in *Nrf1^f/f^;HRGP-Cre* at 1 month old were slightly but statistically significantly less than those in the *Nrf1^f/+^* retina (compare Figure 2M with Figure 2N, and Figure 2U). The number of M-cones decreased to ~10% that of the *Nrf1^f/+^* retinas at 3 months old (compare Figure 2O with Figure 2P) and was almost undetectable at 4 months old (compare Figure 2Q with Figure 2R). No region-specific difference in the number and pattern of the reduction of M-cones was observed (data not shown). Together, these data indicate that the loss of *Nrf1* affects both S- and M-cones in the entire retina and the loss of S-cones in the ventral retinas is slightly faster than that in the dorsal retinas.

### 3.2. Slow and Progressive Reduction of Nrf1-Depleted ipRGC

Next, we examined the effect of *Nrf1*-deleted ipRGCs using *Opn4^Cre^* (Figure 3A). *Opn4^Cre^* is a *Cre* knock-in line in which the expression of *Cre* is under the control of the intrinsic *Opn4* promoter and regulatory regions. *Opn4^Cre^* in conjunction with *Z/EG* reporter has been used to detect all six types of intrinsically photosensitive retinal ganglion cells (ipRGCs) [25,34]. We first confirmed that *Nrf1* is deleted in Opn4-expressing cells by conducting co-immunostaining on 3-month-old *Nrf1^f/+^* and *Nrf1^f/f^*;*Opn4^Cre^* retinas with anti-melanopsin and anti-Nrf1 antibodies (Figure 3B,C). In *Nrf1^f/+^* retina, all melanopsin^+^ ipRGCs express Nrf1 (Figure 3B–B”, *n* = 38). In *Nrf1^f/f^*;*Opn4^Cre^* retina, the number of melanopsin^+^ cells are much less than that of *Nrf1^f/+^* (compared Figure 3B′,C′). Among the 16 melanopsin^+^ ipRGCs examined in *Nrf1^f/f^*;*Opn4^Cre^* retina, 15 lacked Nrf1 staining (Figure 3C′,C”), suggesting that *Nrf1* has been efficiently deleted in *Nrf1^f/f^*;*Opn4^Cre^* retina.

Next, we conducted a longitudinal study to examine how *Nrf1* deletion affects the survival of ipRGCs over time. We immuno-labeled 1-month, 3-month, and 6-month-old *Nrf1^f/+^* and *Nrf1^f/f^*;*Opn4^Cre^* retinas with an anti-melanopsin antibody, then counted melanopsin^+^ ipRGCs. In 1-month-old retinas, the number of melanopsin^+^ cells in *Nrf1^f/+^* and *Nrf1^f/f^*;*Opn4^Cre^* retinas are comparable (Figure 3D,E,J). The number of melanopsin^+^ cells in *Nrf1^f/f^*;*Opn4^Cre^* decreased to ~50% at 3 months old (Figure 3F,G,J), and further dropped to ~20% at 6 months old compared to *Nrf1^f/+^* (Figure 3H–J). These results indicate that the depletion of *Nrf1* caused a slower but progressive ipRGC degeneration, and unlike cones and rods, ipRGCs are more resistant to defective mitochondria biogenesis.

### 3.3. Highly Sensitive Type 2 and 6 Cone Bipolar Cells to Nrf1 Deletion

Next, we examined the *Nrf1*-deleted retinal bipolar cells (BCs), which are a less energy demanding cell type in retinas [35]. We used a *Pcp2-Cre* line to delete *Nrf1* (*Nrf1^f/f^*;*Pcp2-Cre*) (Figure 4A) in retinal bipolar cells. The *Pcp2-Cre* line, when bred with the *Ai9* reporter, has been shown to activate tdTomato mainly in rod BPs as well as type 2 and 6 cone BPs [36]. The *Pcp2-Cre* mouse line also carries a *GFP* reporter gene whose expression in BCs matches the tdTomato expression in *Pcp2-Cre*;*Ai9*. To confirm *Nrf1* deletion in *Pcp2-Cre* cells, we first conducted immunostaining for GFP and Nrf1 expression on *Nrf1^f/+^*;*Pcp2-Cre* (control) and *Nrf1^f/f^*;*Pcp2-Cre* (mutant) retinas at 4 weeks old (Figure 4B,C). In control retinas, all GFP^+^ cells co-express Nrf1 (Figure 4B′), whereas all GFP^+^ cells in mutant retinas lacked Nrf1 (Figure 4C′), indicating that *Pcp2-Cre* is effective in deleting *Nrf1* in bipolar cells.

Next, we compared the number of GFP^+^ BP cells between controls and mutants. At 4 weeks old, the number of GFP^+^ cells in mutant retinal sections was ~50% that of the controls (Figure 4D,E,H). This ratio did not change through 4- to 8-week-old retinas (Figure 4H). However, at 12 weeks old, the number of GFP^+^ cells in mutants dropped to ~10% that of the controls (Figure 4F–H). To distinguish whether rod or cone BPs have different sensitivity to *Nrf1* deletion, we checked GFP co-expression patterns with PKCα, a rod BC marker, and Syt2, which labels type 2 and 6 cone BCs. In 4-week-old *Pcp2-Cre* retinas, we found that GFP^+^PKCα^+^ rod BCs and GFP^+^Syt2^+^ cone BCs account for approximately 58.8% and 23.6%, respectively, of all GFP^+^ cells in the inner nuclear layer (INL). The other 17.6% of PKCα^−^Syt2^−^GFP^+^ cells in INL are likely type 5 and 7 BCs (data not shown). On this basis, we compared GFP co-expression with PKCα and Syt2 at various ages to examine how *Nrf1* deletion affects the survival of different BC subtypes over time. We conducted immunostaining on 4- to 12-week-old control and mutant retinas for rod BPs and types 2 and 6 cone BCs. At 4 weeks old, GFP^+^PKCα^+^ rod BP cells in mutants were ~60% that of the controls (Figure 4I,J,I′,J′,K). The ratio of GFP^+^PKCα^+^ cells between the controls and mutants did not change through 4 to 8 weeks. Eventually, at 12 weeks of age, GFP^+^PKCα^+^ rod BPs drop to ~10% in mutants compared to that of the controls (Figure 4K). In sharp contrast, at 4 weeks, the number of GFP^+^Syt2^+^ cone BPs in mutants dropped to ~20% compared to that of the controls (Figure 4I,J,I”,J”,L). In 8- and 12-week-old mutant retinas, we cannot detect any GFP^+^Syt2^+^ cells. These results revealed a previously unknown cellular property of Syt2^+^ type 2 and 6 cone BCs, i.e., that they are extremely sensitive to defective mitochondria biogenesis compared to rods, cones, RGCs, and rod BCs, suggesting that they may have higher demand for mitochondria-mediated biological functions, or that Nrf1 has a different function in them to support their functions and survival.

### 3.4. The Most Sensitive Genes in Nrf1-Deficient Rod Photoreceptors

Because *Nrf1* deletion in all of the retinal neurons tested led to a similar pattern of progressive or rapid cell death, to understand how neurons respond to *Nrf1* deletion at the transcriptional level, we used rod photoreceptors (PRs) as a model to examine how *Nrf1* deletions affect gene expression. Previously, we found that in rod PR-specific *Nrf1* deletion mice (*Nrf1^f/f^*;*Rho^iCre^*), the number, morphology, and location of mitochondria in rod PRs are abnormal compared to that of the controls; consequently, mitochondria-mediated energy production is impaired and severe rod degeneration takes place, followed by cone degeneration [29].

We selected 4 weeks postnatal age as the experimental time point because the onset of PR degeneration in *Nrf1^f/f^*;*Rho^iCre^* retinas has not started, while the *Nrf1* has been effectively removed by *Rho^iCre^* (Figure 5A) [29]. We reasoned that those genes most sensitive to the loss of *Nrf1* would display greater changes in expression levels. Six 4-week-old *Nrf1^f/+^* (CTL) and *Nrf1^f/f^*;*Rho-iCre* (rod-*Nrf1* deletion) retinas were collected to conduct total RNA-seq.

We first filtered the gene list by adjusting the *p*-value to < 0.05 and identified 1505 differentially expressed genes (DEGs), of which 695 are upregulated and 810 are downregulated in rod PR-specific *Nrf1* mutants (Figure 5B). Next, we selected fold-change ≥ 2 (|(log2FC)| ≥ 1) as the cut-off to reveal *Nrf1*-dependent genes from these 1505 DEGs. By these stringent criteria, we only detected 25 genes whose expression between control and mutant groups was changed (Table 1). To facilitate a meaningful gene ontology analysis, we reduced the cut-off of the fold-change to ≥1.6667 (|(log2FC)| ≥ 0.7369) to define the most sensitive, early responsive *Nrf1*-dependent genes in rod-*Nrf1* deletion. By such a criterion, we identified 58 downregulated and 28 upregulated genes in *Nrf1*-deficient rod PRs (Figure 5C) (Table 1). A heatmap was plotted to show the hierarchical clustering of these 86 DEGs (Figure 5D). We then compared these 86 genes with Nrf1-bound target genes identified in the P0 mouse brain (GSE161808) and found that 57 out of 86 genes are direct targets of Nrf1 (Figure 5C), placing these 57 genes as the early responsive target genes of *Nrf1*-deleted rod PRs (Table 1; gene names in bold). Next, because of the abnormal, mispositioned mitochondria and defective energy production observed in *Nrf1*-deficient PRs, we anticipated that genes involved in various mitochondrial functions would also likely be sensitive to rod-*Nrf1* deletion. We compared the 86 most sensitive genes with gene lists in MitoCarta 2.0 [37,38] and revealed that only 7 genes were involved in mitochondrial functions (Figure 5C) (Table 1; genes marked with asterisk). However, when we used a more lenient criterion by setting the fold-change to ≥1.25 (|(log2FC)| ≥ 0.3219), we recovered 52 mitochondrial genes whose expression was altered in *Nrf1*-deficient rod PRs, and among them, 34 are direct targets of Nrf1 (Table 2; gene names in bold). We color-coded and mapped them in the functional subdomain of the mitochondria (Figure 5E). To quantitatively evaluate RNA-seq data relevant to these genes, we performed qRT-PCR analyses on a subset of these mitochondrial genes and found that these qRT-PCR analyses results were consistent with the RNA-seq data (Figure 5F).

Next, we analyzed the 86 most sensitive genes from the more stringent criteria using gene ontology biological process analysis (GO-BP). The top four categories revealed in GO terms are genes involved in the regulation of double-strand break repair and DNA recombination (Table 3). Interestingly, the genes involved in these processes, including Pot1b, Rad51ap1, Fignl1, Rbbp8, Zcwpw1, Prkdc, Cdc7, Slx4, Rad51ap1, Rbbp8, and Ino80c, are direct targets of Nrf1, suggesting that *Nrf1* deletion in rod PRs led to the misregulation of its direct target genes involved in DNA repair and recombination, conceivably leading to DNA damages in rod photoreceptors. To test whether DNA damage in *Nrf1*-deficient rod PRs precedes the onset of other phenotypes we revealed previously through histology, electron microscopy imaging of mitochondria, and scotopic ERG [29], we conducted TUNEL assay on P21 control and rod-*Nrf1* deletion retinas. Consistent with the transcriptomic data, we detected many TUNEL^+^ rod PRs in P21 rod-*Nrf1* deletion retinas but not in the control retinas (Figure 5G,H), suggesting DNA breakage is the earliest detectable phenotype found in *Nrf1* deletion rod PRs. In 7-week-old rod-*Nrf1* deletion retinas, significantly more TUNEL^+^ cells were detected in rod PRS of rod-*Nrf1* deletion retinas compared to that of control retinas (Figure 5I,J).

## 4. Discussion

In this report, we tested the susceptibility of various retinal neuronal subtypes to defective Nrf1-mediated biological processes including mitochondria biogenesis. We conducted a longitudinal study and tested how quickly retinal neurons degenerate when *Nrf1* is genetically deleted in a cell type-specific manner. It is conceivable that all retinal neurons eventually die when *Nrf1* is removed. However, our data revealed differential susceptibility of different types of retinal neurons to *Nrf1* deletion.

Photoreceptors are known to be an extremely energy-demanding cell type to support several principal energy-consuming functions, such as phototransduction, outer segment disk regeneration, and active ion transport, to repolarize the plasma membrane and re-establish the transmembrane ionic gradient for re-activation [39,40,41]. In *Nrf1*-deleted cone photoreceptors (cone PRs), we found that their tolerance for *Nrf1* deletion is similar to that of rod photoreceptors (rod PRs) [29]. In both cases, PRs degenerate within 4 to 5 months of ages following *Nrf1* deletion, even though cones consume more energy than rods in bright light. In rod-*Nrf1*-mutants, cone PRs also degenerate, owing to the lack of survival factors from RPs [42]. In contrast, in cone-*Nrf1*-mutants, rod PRs remain histologically intact and functional by ERG analysis (data not shown), despite the fact that cone PRs mediate electrical coupling through a rod-cone-rod pathway [43]. It is conceivable that the major cause of PR degeneration in *Nrf1*-mutants is cell-autonomous and due to the defective mitochondrial biogenesis that fails to provide sufficient energy. Furthermore, the defective mitochondrial biogenesis in *Nrf1*-mutant PRs may lead to a defective mitochondrial network, resulting in a barrier for incoming light. A recent study found that mitochondria in cone PRs of ground squirrels may serve as microscopic lenses to help focus light on the photoreceptor pigments. In rod-*Nrf1*-mutants, we have observed misshaped and mispositioned mitochondria in the inner segments of rods. Whether these anomalies in mitochondrial structure and position in *Nrf1*-mutants directly disturb the focus of light and hence cause the decline of ERG activity remains unknown.

Among all GFP^+^ bipolar cells found in the *Pcp2-GFP-Cre* line (58% PKCα^+^ rod bipolar cells, 23% Syt2^+^ type 2 OFF- and type 6 ON-BCs, and 17% PKCa^−^Syt2^−^ BCs that may be type 5 or 7) [36], we found that Syt2^+^ cone BCs are much more sensitive to *Nrf1* deletion than rod BCs and the other subset of cone BCs. At 4 weeks of age, the number of Syt2^+^ cone BCs in *Nrf1^f/f^*;*Pcp2-Cre* retinas has significantly decreased to ~20% compared to that of the *Pcp2-Cre* control, while the number of the other cone BCs and rod BCs only reduced to ~60% at this stage. This observation is intriguing because even in the rod- or cone-specific *Nrf1* mutant retinas, the number and their functionalities characterized by ERG remain similar to that of control littermates at this stage. Most strikingly, in 2-month-old *Nrf1^f/f^*;*Pcp2-Cre* retinas, Syt2^+^ BCs completely disappeared, while other BC types only partially degenerated. This observation reveals a previously unknown character of Syt2^+^ cone BCs: they are extremely sensitive to *Nrf1* deletion and/or defective mitochondrial biogenesis, implicating a high demand for energy and/or mitochondrial biogenesis in this population. In a recent study using the complex-1 component Ndufs4 knockout mouse model [44], death of rod BCs was found taking place at P20, preceding the inflammatory wave that first leads to the death of amacrine cells and then RGCs at P42, suggesting that rod BCs are indeed a sensitive cell type to mitochondrial defect. Our present study supports this notion and further suggests that Syt2^+^ cone BCs are much more sensitive to defective mitochondrial biogenesis than rod BCs and other retinal neuronal types. Among the two Syt2^+^ cone BC subtypes found in *Pcp2-Cre* retinas, it is worth noting that the type-2 OFF BCs possess an interesting cellular property through which they import transcription factor Otx2 from photoreceptors and incorporate it into mitochondria to protect themselves from glutamate excitotoxicity and to support mitochondrial ATP synthesis [45,46]. Whether Nrf1-mediated mitochondrial biogenesis is involved in such a unique intercellular transporting process remains to be studied.

In ganglion cell populations, we previously used a sparse labeling strategy and reported that ~50% of *Nrf1*-deleted Tbr1-expressing OFF RGCs survive for at least 3 months following *Nrf1* deletion [17]. In the present study, we measure the number of melanopsin-expressing ipRGCs and found similar results. Although these two experiments were conducted in different RGC subpopulations using different experimental settings, the similar outcome supports a notion that RGCs are relatively more resistant to defective mitochondrial biogenesis than other retinal neurons. This is rather surprising given that RGCs are most vulnerable to mitochondrial damages because of their unique cellular architecture, including the elaborate dendritic arbors in the inner plexiform layer and the long axons that extend into the brain and connect with other neurons. At the active sites of both regions, the energy demand for producing neurotransmitters, organizing synaptic vesicles, restoring ion gradients, and buffering calcium is extensive. In several human diseases caused by mutations in genes essential for mitochondrial function, including approximately 90% of Leber’s hereditary optic neuropathy cases with point mutations in mitochondria DNA [47] and 75% of autosomal dominant optic atrophy patients with mutations in OPA1 [48,49], RGCs are the most sensitive neurons and die as diseases progress. In the N-methyl-D-aspartic acid-induced excitotoxicity model in retinas, RGCs are also the most sensitive neurons due to the glutamate-evoked rise of calcium mediated by NMDA-type glutamate receptors in RGCs [50,51,52]. A simple explanation is that RGCs may have a backup mechanism temporarily compensating for the loss of *Nrf1*-mediated mitochondrial biogenesis pathways to some extent but not forever. Alternatively, those patients with OPA1 or mitochondrial mutations may also have unrecognized, defective neurons in other parts of the CNS that do not present notable neurological symptoms and pathology [53].

The GO-BP analysis of the 86 most sensitive genes found in *Nrf1*-deficient PRs reveals that the top affected biological processes in gene ontology terms encompass double-strand break repair and DNA recombination. This early response at the transcription level implies that Nrf1 plays a pivotal role in maintaining genome integrity in the central nervous system. Consistent with this idea, Nrf1 has been shown to be physically associated with Prkdc, Xrcc5/Ku70, and Xrcc6 (Ku80) [20], further strengthening Nrf1’s role in maintaining genome integrity at transcriptional and post-transcriptional levels. Additionally, genes with neuroprotective function, such as *Mlf1* and *Bcl2l12*, were upregulated in the *Nrf1*-mutant, likely due to DNA damage and other cellular stresses prior to the onset of degeneration.

When we used a more relaxed criterion in differential gene expression analysis, we found more mitochondrial genes whose expression is modestly affected in *Nrf1*-deleted rod PRs. These genes are different from those found in *Nrf1*-deficient retinal progenitor cells (RPCs) in embryonic retina [29]. For example, expression of several ribosomal or translation-related genes that may be involved in the mitochondrial translation system, including a subset of Mrpl genes and Rpl10a, Rpusd3, Sars2, Mtg2, and Vars2, were altered in *Nrf1*-deficient rod PRs but not in *Nrf1*-deficient RPCs. 8-Oxoguanine (8-oxoG) is one of the major base lesions in oxidative stressed DNA [54]. 8-oxoG can pair with adenine and cytosine, leading to base-substitution mutagenesis [55]. Mutyh, an adenine DNA glycosylase, removes misincorporated adenine in template DNA that pairs with 8-oxoG in the opposite strand [56]. The upregulation of *Mutyh* in rod-*Nrf1*-deleted retinas supports the notion that the damaged PRs activate DNA repair machinery. Hexokinase 1 (HK1), which functions at the first step of the glycolysis pathway, was upregulated in *Nrf1*-deficient rods but downregulated in *Nrf1*-deficient RPCs, suggesting that *Nrf1* deletion affects metabolic pathways differently in neural development and degeneration states. Additionally, photoreceptors are known to be glycolytic [57,58,59]. We also found that the expression of 6 genes involved in glycolysis pathway were modestly affected in rod-*Nrf1*-deleted retinas (Table 4), suggesting that rod-specific *Nrf1* deletion moderately affects glycolysis and mitochondrial energy production pathways. In addition, RNA-seq revealed a subset of misregulated genes involved in neurodegeneration in CNS. For example, Bardet-Biedl syndrome 2 protein homolog *Bbs2* is downregulated by ~0.5 fold in rod-*Nrf1*-deleted retinas. It has been shown that *Bbs2* mutation causes photoreceptor degeneration with mislocalized rhodopsin [60]. *Ska3*, an important kinetochore component involved in proper chromosome segregation during mitosis [61], is upregulated in the *Nrf1*-mutant to ~1.8 fold. It has been shown that *Ska3* upregulation activates PI3K/Akt signaling pathway, which promotes cell proliferation in cervical cancer [62]. Whether the upregulation of Ska3 in *Nrf1*-deleted photoreceptor leads to aberrant cell cycle re-entry and cell death remains unclear [63].

Overall, our findings offer a deeper understanding of how defective Nrf1-mediated mitochondrial biogenesis might contribute to the pathology and progression of neurodegeneration in the retina, and suggest that a fine-tuned Nrf1 expression level is essential for safeguarding genetic fidelity indispensable for the long life of mature neurons.

## Figures and Tables

**Figure 1 cells-11-02203-f001:**
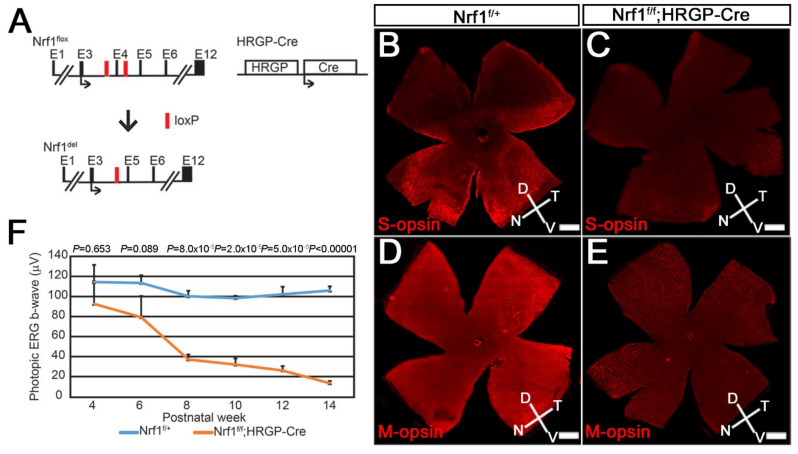
*Nrf1* conditional knockout by *HRGP-Cre* causes impaired cone photoreceptor function. (**A**) Schematic diagram showing *Nrf1^flox^* allele, *Nrf1*-deleted allele and *HRGP-Cre* allele. Exons are indicated as E1–E12. Black arrows indicate the translational start sites for Nrf1 and Cre protein. Red boxes indicate loxP sites. (**B**–**E**) Immunostaining of 3-month-old *Nrf1^f/+^* (**B**,**D**) and *Nrf1^f/f^;HRGP-Cre* (**C**,**E**) flatmount retinas showing the expression of S-opsin (**B**,**C**) and M-opsin (**D**,**E**). (**F**) ERG of *Nrf1^f/+^* and *Nrf1^f/f^;HRGP-Cre* mice under light-adapted conditions. Scale bars: 500 μm.

**Figure 2 cells-11-02203-f002:**
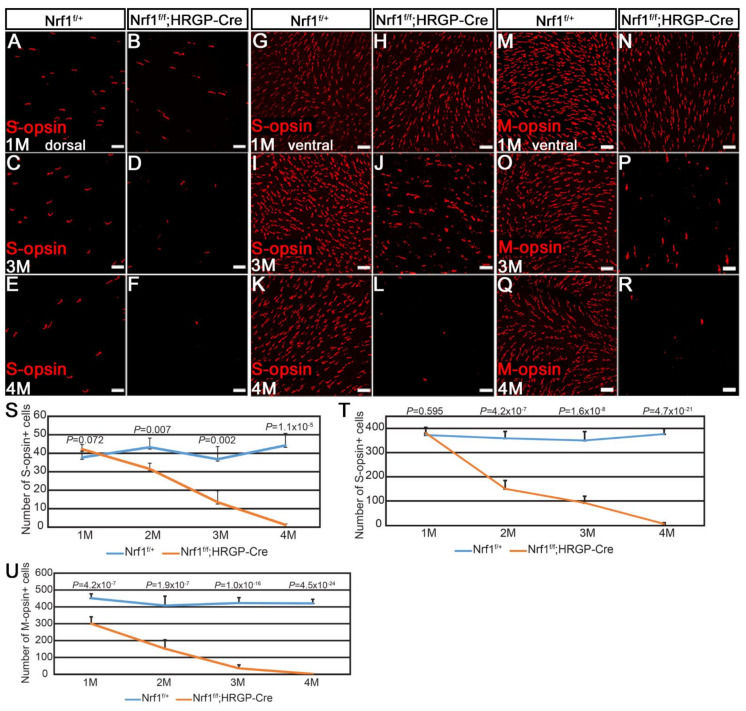
*Nrf1* conditional knockout by *HRGP-Cre* causes degeneration of S- and M-opsin. (**A**–**R**) Immunostaining of *Nrf1^f/+^* (**A**,**C**,**E**,**G**,**I**,**K**,**M**,**O**,**Q**) and *Nrf1^f/f^;HRGP-Cre* (**B**,**D**,**F**,**H**,**J**,**L**,**N**,**P**,**R**) retinal flat-mounts with anti-B opsin (**A**–**L**) and anti-RG opsin (**M**–**R**) at 1 month, 3 months, and 4 months old. (**S**,**T**) Number of S-opsin^+^ cells at dorsal area (**S**) and ventral area (**T**) of *Nrf1^f/+^* and *Nrf1^f/f^;HRGP-Cre* retinal flatmount. (**U**) Number of M-opsin^+^ cells of *Nrf1^f/+^* and *Nrf1^f/f^;HRGP-Cre* retinal flatmount. Scale bars: 20 μm.

**Figure 3 cells-11-02203-f003:**
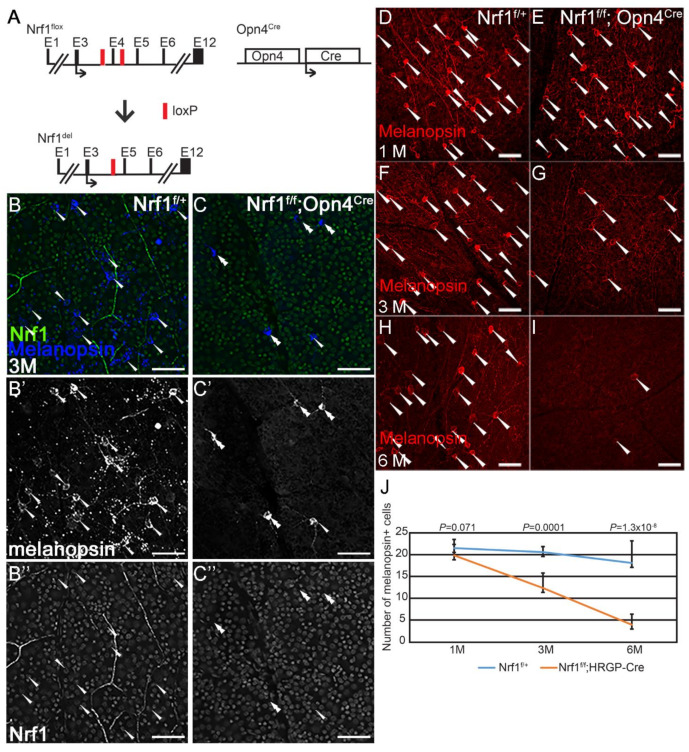
*Nrf1* conditional deletion by *Opn4^cre^* causes slow degeneration of ipRGCs. (**A**) Schematic representation of *Nrf1^flox^* allele, *Nrf1*-deleted allele, and *Opn4^Cre^* allele. Exons are indicated as E1–E12. Black arrows indicate the translational start sites for Nrf1 and Cre protein. Red boxes indicate loxP sites. (**B**,**C**) Immunostaining of 3-month-old *Nrf1^f/+^* (**B**) and *Nrf1^f/f^*;*Opn4^cre^* (**C**) retinal flat-mounts with anti-melanopsin (**B′**,**C′**) and anti-Nrf1 (**B″**,**C″**) antibodies. (**B**,**C**) merged images. Melanopsin^+^ and Nrf1^+^ cells are indicated with arrowheads. Melanopsin^+^ Nrf1^−^ cells are indicated with double arrowheads. (**D**–**I**) Immunostaining of *Nrf1^f/+^* (**D**,**F**,**H**) and *Nrf1^f/f^*;*Opn4^cre^* (**E**,**G**,**I**) retinal flat-mounts with anti-melanopsin antibody. Melanopsin^+^ cells are indicated with arrowhead. (**J**) Number of melanopsin^+^ cells of *Nrf1^f/+^* and *Nrf1^f/f^*;*Opn4^cre^* retinal flat-mounts at 1 month, 3 months, and 6 months old. Scale bars: 50 μm.

**Figure 4 cells-11-02203-f004:**
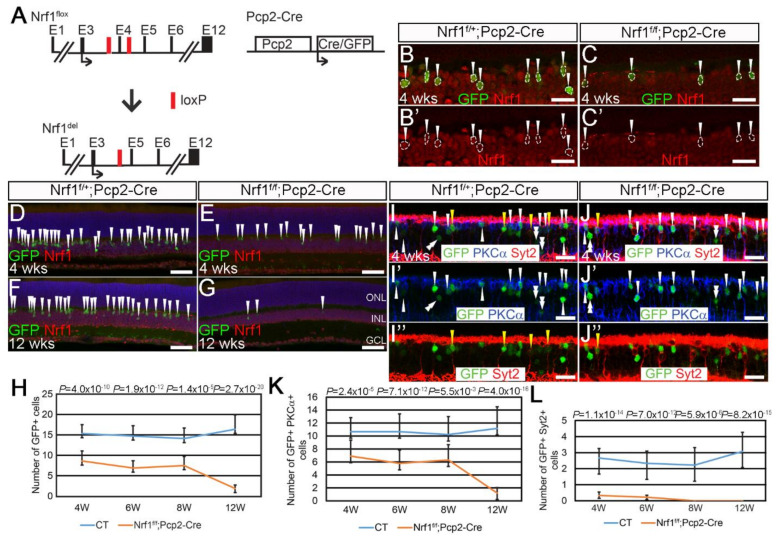
*Nrf1* conditional deletion by *Pcp2-Cre* causes rapid degeneration of type 2 and 6 cone bipolar cells. (**A**) Schematic illustration of *Nrf1^flox^*, *Nrf1^del^*, and *Pcp2-Cre* alleles. Exons are indicated as E1–E12. Black arrows indicate the translational start sites for Nrf1 and Cre protein. Red boxes indicate loxP sites. (**B**,**C**) Immunostaining of 4-week-old *Nrf1^f/+^*;*Pcp2-Cre* (**B**,**B′**) and *Nrf1^f/f^*;*Pcp2-Cre* (**C**,**C′**) retinas with anti-GFP (**B**,**C**) and anti-Nrf1 (**B**,**B′**,**C**,**C′**) antibodies. GFP^+^ cells are indicated with arrowhead. (**D**–**G**) Immunostaining of 4- and 12-week-old *Nrf1^f/+^*;*Pcp2-Cre* (**D**,**F**) and *Nrf1^f/f^*;*Pcp2-Cre* (**E**,**G**) retinas with anti-GFP and anti-Nrf1 antibodies. GFP^+^ cells are indicated with arrowheads. (**H**) Number of GFP+ cells of *Nrf1^f/+^*;*Pcp2-Cre* and *Nrf1^f/f^*;*Pcp2-Cre* retinas at 4, 6, 8, and 12 weeks old. (**I**,**J**) Immunostaining of *Nrf1^f/+^*;*Pcp2-Cre* (**I**,**I′**,**I″**) and *Nrf1^f/f^*;*Pcp2-Cre* (**J**,**J′**,**J″**) retinas at 4 weeks old with anti-GFP (**I**–**I″**,**J**–**J″**), anti-PKCα (**I**,**I′**,**J**,**J′**), and anti-Syt2 (**I**,**I″**,**J**,**J″**) antibodies. GFP^+^PKCα^+^ cells, GFP^+^Syt2^+^ cells, and GFP^+^PKCα^−^Syt2^−^ cells are indicated with white arrowheads, yellow arrowheads, and white double-arrowheads, respectively. (**K**) Number of GFP^+^PKCα^+^ cells of *Nrf1^f/+^*;*Pcp2-Cre* and *Nrf1^f/f^*;*Pcp2-Cre* retinas at 4, 6, 8, and 12 weeks old. (**L**) Number of GFP^+^Syt2^+^ cells of *Nrf1^f/+^*;*Pcp2-Cre* and *Nrf1^f/f^*;*Pcp2-Cre* retinas at 4, 6, 8, and 12 weeks old. Scale bars: 20 μm in (**B**,**B′**,**C**,**C′**,**I**,**I′**,**I″**,**J**,**J′**,**J″**) and 50 μm in (**D**–**G**).

**Figure 5 cells-11-02203-f005:**
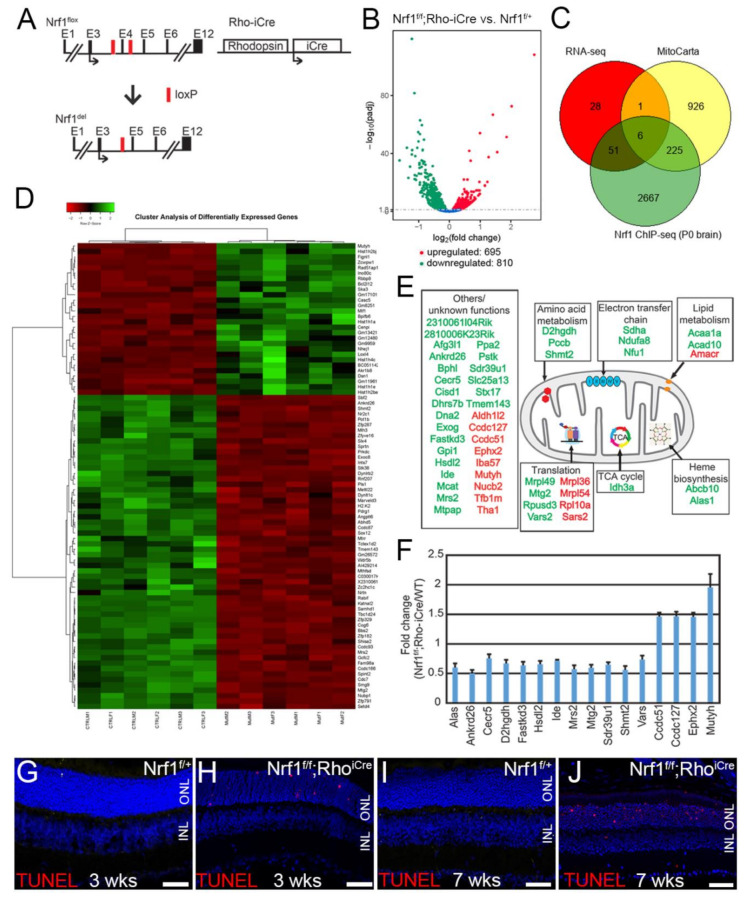
*Nrf1* conditional deletion by *Rho^iCre^* causes dynamic gene expression alteration before the onset of photoreceptor degeneration. (**A**) Schematic representation of *Nrf1^flox^* allele, *Nrf1* deleted allele, and *Rho^iCre^* allele. Exons are indicated as E1–E12. Black arrows indicate the translational start sites for Nrf1 and iCre protein. Red boxes indicate loxP sites. (**B**) Volcano plot showing the differentially expressed genes. The *x*-axis indicates the log_2_ fold change of genes between mutant and control. The *y*-axis indicates the −log_10_(adjusted *p*-value). Each dot represents a gene. Blue dots indicate no significant difference; red dots indicate upregulated DEGs; green dots indicate downregulated DEGs. (**C**) Venn diagram showing coexpressed genes among the three gene lists described in the text. (**D**) Hierarchical clustering heatmap showing the 86 most sensitive, early responsive DEGs in rod-specific *Nrf1*-deleted retinas. (**E**) Schematic mapping of mitochondrion and functional annotation of upregulated (red) and downregulated (green) genes in *Nrf1^f/f^*;*Rho^iCre^* retinas detected by RNA-seq. (**F**) qRT-PCR analysis of *wild-type* and *Nrf1^f/f^*;*Rho^iCre^* retinas, confirming the changes of mRNA levels detected by RNA-seq. (**G**–**J**) TUNEL assay of 3- and 7-week-old *Nrf1^f/+^* (**G**,**I**) and *Nrf1^f/f^*;*Rho^iCre^* (**H**,**J**) retinas. ONL: outer nuclear layer. INL: inner nuclear layer. Scale bars: 50 μm.

**Table 1 cells-11-02203-t001:** The most sensitive genes in *Nrf1^f/f^*;*Rho^iCre^* retina. Gene names in bold font are direct targets of Nrf1 found in P0 brain (GEO dataset: GSE161808). Genes marked with an asterisk encode proteins found in mitochondria.

Gene Name	Description	Log_2_ Fold Change (Nrf1^f/f^;Rho^iCre^/Nrf1^f/+^)	Adjusted *p* Value
**Ccdc87**	coiled-coil domain containing 87	−1.6234	5.89 × 10^−36^
Rnf207	ring finger protein 207	−1.3834	1.10 × 10^−44^
**AI429214**	expressed sequence AI429214	−1.3369	6.84 × 10^−24^
C030017K20Rik	RIKEN cDNA C030017K20 gene	−1.2807	1.19 × 10^−31^
**Zfp182**	zinc finger protein 182	−1.2239	4.54 × 10^−40^
Tbc1d24	TBC1 domain family member 24	−1.2234	9.22 × 10^−120^
**Cdc7**	cell division cycle 7 (*S. cerevisiae*)	−1.2132	4.45 × 10^−43^
Samhd1	SAM domain and HD domain 1	−1.139	2.41 × 10^−82^
Zc2hc1c	zinc finger C2HC-type containing 1C	−1.1341	4.26 × 10^−23^
**Wdr5b**	WD repeat domain 5B	−1.1292	8.21 × 10^−26^
**Sprtn**	SprT-like N-terminal domain	−1.1122	6.12 × 10^−39^
H2-K2	histocompatibility 2 K region locus 2	−1.0906	1.58 × 10^−22^
**Shisa2**	shisa family member 2	−1.0698	2.44 × 10^−25^
**Mlh3**	mutL homolog 3 (*E. coli*)	−1.0473	1.15 × 10^−44^
Ankrd26 *	ankyrin repeat domain 26	−1.0158	1.68 × 10^−55^
**Cog6**	component of oligomeric golgi complex 6	−1.0071	4.78 × 10^−49^
**Zfp287**	zinc finger protein 287	−0.99342	7.33 × 10^−27^
**Mtg2 ***	mitochondrial ribosome associated GTPase 2	−0.97742	5.34 × 10^−33^
**Katnal2**	katanin p60 subunit A-like 2	−0.97416	3.88 × 10^−34^
**Mthfsd**	methenyltetrahydrofolate synthetase domain containing	−0.97072	8.13 × 10^−31^
Zfp791	zinc finger protein 791	−0.96542	3.25 × 10^−19^
**Zfp329**	zinc finger protein 329	−0.9606	4.01 × 10^−54^
**Pot1b**	protection of telomeres 1B	−0.95894	9.62 × 10^−21^
Bbs2	Bardet-Biedl syndrome 2 (human)	−0.95414	1.22 × 10^−63^
**Dynlrb2**	dynein light chain roadblock-type 2	−0.94971	6.89 × 10^−14^
**Abhd5**	abhydrolase domain containing 5	−0.9241	1.01 × 10^−26^
**Marveld3**	MARVEL (membrane-associating) domain containing 3	−0.91459	6.08 × 10^−15^
**Ccdc166**	coiled-coil domain containing 166	−0.90586	1.81 × 10^−31^
Exoc8	exocyst complex component 8	−0.90494	6.33 × 10^−36^
**Stk38**	serine/threonine kinase 38	−0.89765	2.81 × 10^−60^
**Tctex1d2**	Tctex1 domain containing 2	−0.89357	9.03 × 10^−14^
Gm26572	predicted gene 26572	−0.88514	1.06 × 10^−16^
**Rabif**	RAB interacting factor	−0.87374	6.53 × 10^−43^
**Dynlt1c**	dynein light chain Tctex-type 1C	−0.87284	3.64 × 10^−11^
**Spint2**	serine protease inhibitor Kunitz type 2	−0.87063	2.51 × 10^−43^
**Fam98a**	family with sequence similarity 98 member A	−0.86343	2.20 × 10^−35^
**Slx4**	SLX4 structure-specific endonuclease subunit homolog (*S. cerevisiae*)	−0.85413	9.39 × 10^−34^
**Mtrr**	5-methyltetrahydrofolate-homocysteine methyltransferase reductase	−0.84843	1.96 × 10^−18^
**Setd4**	SET domain containing 4	−0.83398	1.40 × 10^−17^
**Gcfc2**	GC-rich sequence DNA binding factor 2	−0.83387	2.49 × 10^−15^
**Sox12**	SRY (sex determining region Y)-box 12	−0.82933	7.00 × 10^−21^
Sbf2	SET binding factor 2	−0.82821	2.80 × 10^−50^
**Nubp1**	nucleotide binding protein 1	−0.82325	3.47 × 10^−19^
**Angptl6**	angiopoietin-like 6	−0.82273	9.67 × 10^−13^
**Mrs2 ***	MRS2 magnesium homeostasis factor homolog (*S. cerevisiae*)	−0.81398	6.57 × 10^−35^
**Pdrg1**	p53 and DNA damage regulated 1	−0.79343	1.45 × 10^−27^
Zfyve16	zinc finger FYVE domain containing 16	−0.78442	1.04 × 10^−24^
**Ints7**	integrator complex subunit 7	−0.783	5.35 × 10^−33^
**Shmt2 ***	serine hydroxymethyltransferase 2 (mitochondrial)	−0.78075	2.29 × 10^−29^
**Mettl22**	methyltransferase like 22	−0.77958	2.18 × 10^−16^
**Ccdc93**	coiled-coil domain containing 93	−0.77053	2.11 × 10^−27^
**Smg9**	smg-9 homolog nonsense mediated mRNA decay factor (*C. elegans*)	−0.76729	3.83 × 10^−24^
**Tmem143 ***	transmembrane protein 143	−0.75985	4.64 × 10^−20^
**Prkdc**	protein kinase DNA activated catalytic polypeptide	−0.75349	1.45 × 10^−18^
**2310061I04Rik ***	RIKEN cDNA 2310061I04 gene	−0.75092	6.93 × 10^−17^
**Pls1**	plastin 1 (I-isoform)	−0.74708	2.22 × 10^−10^
**Nrtn**	neurturin	−0.74655	2.40 × 10^−12^
**Nr2c1**	nuclear receptor subfamily 2 group C member 1	−0.74308	2.67 × 10^−19^
Hist1h1a	histone cluster 1 H1a	0.74952	7.45 × 10^−8^
**Gm13421**	predicted gene 13421	0.77879	2.71 × 10^−14^
Hist1h4c	histone cluster 1 H4c	0.78665	1.79 × 10^−8^
Bpifb6	BPI fold containing family B member 6	0.79491	2.04 × 10^−8^
Hist1h2be	histone cluster 1 H2be	0.7962	2.31 × 10^−14^
**Nhej1**	nonhomologous end-joining factor 1	0.80335	1.01 × 10^−8^
BC051142	cDNA sequence BC051142	0.81214	1.83 ×10^−10^
Hist1h2bj	histone cluster 1 H2bj	0.83377	3.03 ×10^−9^
**Ska3**	spindle and kinetochore associated complex subunit 3	0.84721	1.54 ×10^−9^
Gm11961	predicted gene 11961	0.87529	2.79 × 10^−15^
**Hist1h1e**	histone cluster 1 H1e	0.93176	2.49 × 10^−15^
Gm9959	predicted gene 9959	0.93908	1.03 × 10^−11^
**Rbbp8**	retinoblastoma binding protein 8	0.97505	3.74 × 10^−21^
Mlf1	myeloid leukemia factor 1	0.98101	3.09 × 10^−15^
Bcl2l12	BCL2-like 12 (proline rich)	0.98105	9.18 × 10^−13^
Gm17101	predicted gene 17101	0.98366	7.78 × 10^−13^
Gm12480	predicted gene 12480	0.98839	5.94 × 10^−13^
**Ino80c**	INO80 complex subunit C	0.99506	8.83 × 10^−55^
Gm8251	predicted gene 8251	1.1522	1.03 × 10^−18^
**Dsn1**	DSN1 MIND kinetochore complex component homolog (*S. cerevisiae*)	1.1532	1.85 × 10^−18^
Loxl4	lysyl oxidase-like 4	1.2333	3.80 × 10^−21^
**Mutyh ***	mutY homolog (*E. coli*)	1.2731	3.28 × 10^−38^
**Zcwpw1**	zinc finger CW type with PWWP domain 1	1.4136	1.66 × 10^−67^
**Rad51ap1**	RAD51 associated protein 1	1.5557	1.07 × 10^−41^
**Cenpi**	centromere protein I	1.8589	4.28 × 10^−52^
**Fignl1**	fidgetin-like 1	2.0243	2.37 × 10^−73^
**Casc5**	cancer susceptibility candidate 5	2.7592	6.88 × 10^−109^

**Table 2 cells-11-02203-t002:** Mitochondrial genes with modest changed expression levels in *Nrf1^f/f^*;*Rho^iCre^* retina. Gene names in bold font are direct targets of *Nrf1* found in P0 brain (GEO dataset: GSE161808).

Gene Name	Description	Log_2_ Fold change (Nrf1^f/f^;Rho^iCre^/Nrf1^f/+^)	Adjusted *p* Value
Ankrd26	ankyrin repeat domain-containing protein 26	−1.0158	1.68 × 10^−55^
**Mtg2**	mitochondrial ribosome associated GTPase 2	−0.97742	5.34 × 10^−33^
**Mrs2**	MRS2 magnesium homeostasis factor homolog (*S. cerevisiae*)	−0.81398	6.57 × 10^−35^
**Shmt2**	serine hydroxymethyltransferase 2 (mitochondrial)	−0.78075	2.29 × 10^−29^
**Tmem143**	transmembrane protein 143	−0.75985	4.64 × 10^−20^
**2310061I04Rik**	RIKEN cDNA 2310061I04 gene	−0.75092	4.12 × 10^−19^
Fastkd3	FAST kinase domains 3	−0.73257	6.15 × 10^−12^
**Pstk**	phosphoseryl tRNA kinase	−0.73031	1.28 × 10^−16^
Alas1	aminolevulinic acid synthase 1	−0.70115	9.42 × 10^−25^
**Hsdl2**	hydroxysteroid dehydrogenase like 2	−0.59599	1.75 × 10^−20^
**D2hgdh**	D2 hydroxyglutarate dehydrogenase	−0.58842	1.56 × 10^−11^
**Sdr39u1**	short chain dehydrogenase/reductase family 39U member 1	−0.55177	1.20 × 10^−14^
Sdha	succinate dehydrogenase complex subunit A flavoprotein (Fp)	−0.54497	5.27 × 10^−32^
Cecr5	cat eye syndrome chromosome region candidate 5	−0.53991	1.65 × 10^−11^
Slc25a13	solute carrier family 25 (mitochondrial carrier adenine nucleotide translocator) member 13	−0.53462	4.91 × 10^−4^
Acad10	acyl Coenzyme A dehydrogenase family member 10	−0.53414	2.61 × 10^−6^
**Pccb**	propionyl Coenzyme A carboxylase beta polypeptide	−0.52005	2.09 ×10^−15^
**Exog**	endo/exonuclease (5′-3′) endonuclease G-like	−0.51228	1.76 × 10^−9^
Ide	insulin degrading enzyme	−0.50673	1.97 × 10^−22^
**Mcat**	malonyl CoA:ACP acyltransferase (mitochondrial)	−0.48803	1.60 × 10^−7^
**Afg3l1**	AFG3-like AAA ATPase 1	−0.48699	6.23 × 10^−16^
2810006K23Rik	RIKEN cDNA 2810006K23 gene	−0.47985	2.65 × 10^−7^
**Vars2**	valyl-tRNA synthetase 2 mitochondrial (putative)	−0.45605	9.55 × 10^−7^
Mtpap	mitochondrial poly(A)_polymerase	−0.45337	1.37 × 10^−5^
**Dna2**	DNA replication helicase 2 homolog (yeast)	−0.44603	0.003128
**Mrpl49**	mitochondrial ribosomal protein L49	−0.43346	1.64 × 10^−8^
**Nfu1**	NFU1 iron sulfur cluster scaffold homolog (*S. cerevisiae*)	−0.42766	7.19 × 10^−9^
**Rpusd3**	RNA pseudouridylate synthase domain containing 3	−0.41583	1.73 × 10^−4^
**Abcb10**	ATP-binding cassette subfamily B (MDR/TAP) member 10	−0.41548	3.37 × 10^−12^
Ppa2	pyrophosphatase (inorganic) 2	−0.40886	9.23 × 10^−6^
**Bphl**	biphenyl hydrolase like (serine hydrolase breast epithelial mucin associated antigen)	−0.39912	9.78 × 10^−4^
**Dhrs7b**	dehydrogenase/reductase (SDR family) member 7B	−0.39831	3.88 × 10^−6^
**Cisd1**	CDGSH iron sulfur domain 1	−0.3853	5.47 × 10^−6^
**Stx17**	syntaxin 17	−0.38146	4.69 × 10^−7^
Gpi1	glucose phosphate isomerase 1	−0.35742	2.27 × 10^−6^
**Ndufa8**	NADH dehydrogenase (ubiquinone) 1 alpha subcomplex 8	−0.34492	2.18 × 10^−4^
**Acaa1a**	acetyl-Coenzyme A acyltransferase 1A	−0.33944	2.45 × 10^−9^
**Idh3a**	isocitrate dehydrogenase 3 (NAD+) alpha	−0.32204	5.08 × 10^−9^
**Iba57**	IBA57 iron sulfur cluster assembly homolog (*S. cerevisiae*)	0.33473	2.31 × 10^−4^
Tha1	threonine aldolase 1	0.34796	4.24 × 10^−3^
**Amacr**	alpha methylacyl CoA racemase	0.36987	1.82 × 10^−3^
Aldh1l2	aldehyde dehydrogenase 1 family member L2	0.37356	3.79 × 10^−3^
Tfb1m	transcription factor B1 mitochondrial	0.37658	4.31 × 10^−3^
**Mrpl54**	mitochondrial ribosomal protein L54	0.38391	2.29 × 10^−4^
**Mrpl36**	mitochondrial ribosomal protein L36	0.41616	9.30 × 10^−5^
Nucb2	nucleobindin 2	0.48404	4.16 × 10^−16^
**Sars2**	seryl-aminoacyl-tRNA synthetase 2	0.54314	2.87 × 10^−10^
Ephx2	epoxide hydrolase 2 cytoplasmic	0.56004	1.71 × 10^−14^
Ccdc51	coiled-coil domain containing 51	0.57846	1.49 × 10^−7^
**Ccdc127**	coiled-coil domain containing 127	0.58535	1.02 × 10^−22^
**Rpl10a**	ribosomal protein L10A	0.64403	1.49 × 10^−42^
**Mutyh**	mutY homolog (*E. coli*)	1.2731	3.28 × 10^−38^

**Table 3 cells-11-02203-t003:** Top four GO terms from analysis of the 86 most sensitive genes in *Nrf1^f/f^*;*Rho^iCre^* retina.

GO Biological Process ID	Number of Focused Genes	Fold Enrichment	Raw *p*-Value	FDR	Gene Name
**regulation of double-strand break repair** (**GO:2000779**)	6	18.07	1.35 × 10^−6^	3.05 × 10^−3^	**Pot1b, Rad51ap1, Fignl1, Rbbp8, Zcwpw1, Prkdc**
**double-strand break repair via homologous recombination** (**GO:0000724**)	5	16.42	1.69 × 10^−5^	1.91 × 10^−2^	**Cdc7, Slx4**, Samhd1, **Rad51ap1, Rbbp8**
**regulation of DNA recombination** (**GO:0000018**)	6	15.04	3.75 × 10^−6^	6.57 × 10^−3^	**Pot1b, Rad51ap1, Fignl1, Rbbp8, Zcwpw1**, Hist1h1a
**positive regulation of DNA repair** (**GO:0045739**)	5	14.79	2.74 × 10^−5^	2.70 × 10^−2^	**Rad51ap1, Rbbp8, Zcwpw1, Ino80c, Prkdc**

**Table 4 cells-11-02203-t004:** Genes involved in glycolysis pathway with altered expression in *Nrf1^f/f^*;*Rho^iCre^* retina.

Gene Name	Description	Fold Change (Nrf1^f/f^;Rho^iCre^/Nrf1^f/+^)	Adjusted *p* Value
**Hk1**	Hexokinase 1	1.24	5.35 × 10^−10^
**Gpi1**	Glucose-6-phsphate isomerase 1	0.78	2.27 × 10^−6^
**Pgam1**	Phosphoglycerate mutase 1	0.81	1.57 × 10^−4^
**Pgm2**	Phosphoglucomutase-2	0.80	1.34 × 10^−3^
**Eno1**	Enolase 1, alpha non-neuron	0.82	5.76 × 10^−3^
**Pkm**	Pyruvate kinase, muscle	0.80	5.76 × 10^−6^

## Data Availability

RNA-seq data described in this paper have been deposited in the GEO database (GSE150258).

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
