# Peer review of "Differential Susceptibility of Retinal Neurons to the Loss of Mitochondrial Biogenesis Factor Nrf1"

_cells, 2022, doi:10.3390/cells11142203_

Round 1

Reviewer 1 Report

This paper by Kiyama et al explores how Cre recombinase-mediated loss of nuclear respiratory factor 1 (Nrf1) leads to cell death of different neuronal cells of the retina. Overall, the immunofluorescence images provided seem convincing and well done, whereas the RNAseq experiments seem to be a bit more of a dumping ground for data. The absence of including appropriate controls, however make the data difficult to evaluate absolutely.

The major issue that I have with the paper is the lack of control animals expressing only the indicated recombinase. Cre lines are notorious for causing untoward effects in postmitotic cells, especially in the eye. Thus, utilization of inducible Cre’s can result in higher degrees of experimental consistency with reduced off target effects. While original publications describing the development of these Cre driver lines may have provided details supporting the notion of no detrimental effects due to Cre expression (i.e., neuron or cell loss), the authors should perform their due diligence where in their own studies.

Along these lines, how do we know that the GO terms listed in Table 3 (DNA damage and repair) are not simply due to Cre recombinase?   Having a Cre only line here would demonstrate what effects are really due to Nrf1 loss vs. Cre presence and function as a DNA recombinase.

For the lay audience, are these Cre lines transgenics or knockins? How consistent is the Cre expression? Is it mosaic? while the answers to these questions may not change the overall perception of this paper, it is important information for the reader to know for evaluation purposes.

Characterizing promoter strength and timing would help the reader understand which effects might be due to inherent susceptibility of the neurons to death vs. delayed recombination.

If scRNAseq could be performed, it would provide more targeted results. Additionally, it would be good to try to portray the RNAseq data better, instead of a 3.5 page table.

Having some metric that it consistent across all the models would give a better gauge of similarities vs. differences in the overall retinal effects of loss of Nrf1 in different cells. Even something simple like H&E histology could provide some additional comparative power.

Author Response

Please see attached PDF file.

Reviewer 2 Report

This study investigated how different retinal neurons responded to the loss of Nrf1provding evidence that the disruption of Nrf1-mediated mitochondrial biogenesis results in a slow, progressive degeneration of all retinal cell types examined. Transcriptome analysis on 26 rod specific Nrf1 deletion uncovered a previously unknown role of Nrf1 in maintaining genome stability. Overall, the manuscript is well written and neatly presented. The findings are interesting and provide new information for ophthalmology researchers. The quality of the manuscript can be further improved based on the following suggestions.

Only photopic ERG data is provided, explain reason for not conducting scotopic ERG for assessing possible changes in certain scotopic ERG components. Although it stated that rod ERG ‘data not shown,’ the reviewer is curious to know if all different components of the ERG waveform was properly assessed to make this conclusion

Fig 1- Figure legend is wrongly labelled (B is not ERG). ERG P value not given

Statistical Analysis: It is stated that “Results were considered significant when 142 P <0.05” however in figure legends it is stated as ‘P=xxx’, please correct this

ERG P value not given for Fig 1

P value not given for Fig 3J

Fig3A and Fig4 A-Providing better labeling or more clear explanation in the figure legend is desirable

Providing a list of antibodies used along with sufficient information is desirable

Author Response

Please see attached PDF file.
